# Paracrine communication maximizes cellular response fidelity in wound signaling

L Naomi Handly[1], Anna Pilko[1], Roy Wollman[1,2,3]*

[1]Department of Chemistry and Biochemistry, University of California, San Diego, La Jolla, United States; [2]Section for Cell and Developmental Biology , Division of Biological Sciences, University of California, San Diego, La Jolla, United States; [3]San Diego Center for Systems Biology, La Jolla, United States

**Abstract** Population averaging due to paracrine communication can arbitrarily reduce cellular response variability. Yet, variability is ubiquitously observed, suggesting limits to paracrine averaging. It remains unclear whether and how biological systems may be affected by such limits of paracrine signaling. To address this question, we quantify the signal and noise of $Ca^{2+}$ and ERK spatial gradients in response to an in vitro wound within a novel microfluidics-based device. We find that while paracrine communication reduces gradient noise, it also reduces the gradient magnitude. Accordingly we predict the existence of a maximum gradient signal to noise ratio. Direct in vitro measurement of paracrine communication verifies these predictions and reveals that cells utilize optimal levels of paracrine signaling to maximize the accuracy of gradient-based positional information. Our results demonstrate the limits of population averaging and show the inherent tradeoff in utilizing paracrine communication to regulate cellular response fidelity.

*For correspondence: rwollman@ucsd.edu

**Competing interests:** The authors declare that no competing interests exist.

## Introduction

Cellular variability is likely a biological trait with significant phenotypic consequences. Technological advances in single-cell measurement methodologies reveal substantial cellular variability. For instance, single-cell quantification of protein concentration variability between cells shows that the concentration of many signaling molecules can vary by ~25% (coefficient of variation) (*Sigal et al., 2006*; *Bar-Even et al., 2006*; *Spencer et al., 2009*). Furthermore, a large and rapidly growing body of single-cell transcriptomics experiments further demonstrates that cells homogeneous in 'type' have substantially heterogeneous gene expression patterns (*Junker and Van Oudenaarden, 2014*). The origin of this cellular variability has been traced to fundamental properties of gene expression. Notably, single-molecule kinetics regulates gene expression and, as a result, is an inherently stochastic process (*Sanchez and Golding, 2013*).

While the costs and benefits of cellular variability are likely dependent on the specific physiological context, the functional significance of cellular variability suggests that cellular variability magnitude is regulated. Functional analysis of cellular response variability demonstrates that the observed cellular variability affects the core function of signaling networks. Despite a homogenous environment, cells respond in a heterogeneous manner due to biological variability. Response variability potentially degrades transmitted information and decreases downstream effector ability to reliably respond to environmental changes (*Selimkhanov et al., 2014*; *Cheong et al., 2011*; *Voliotis et al., 2014*; *Hansen and O'Shea, 2015*). The abundance of cellular variability throughout biological processes and the potential consequences of information degradation suggest that biological systems have developed mechanisms to regulate cellular variability. However, cellular variability is not

**eLife digest** The human body is made up of many different types of cell that are each specialized to perform particular roles. Although each cell type has the same set of genes, the level of activity (or "expression") of these genes varies between each type. Additionally, gene expression in cells of the same type can vary due to randomness in the regulation of genes.

Although variation in gene expression between cells can allow populations of cells to adapt to a changing environment, variability can also cause problems when many different cells need to work together. A system called "paracrine signaling" allows cells to communicate with each other by releasing signaling molecules that bind to and activate surrounding cells. The distance that this molecule travels, or the "paracrine communication distance", determines how many surrounding cells each cell can communicate with to coordinate their responses. However, it is not clear what impact paracrine signaling has on the variability between cells, or what limitations there are on the size of the paracrine communication distance.

Cells that are damaged during wounding immediately release a molecule called ATP, which acts as a danger signal to activate the wound healing process in the surrounding cells. The release of ATP from wounded cells forms a spatial gradient in the surrounding healthy cells and stimulates the release of molecules called growth factors that are required for the healing process.

Here, Handly et al. developed a new device to study the responses of human cells to a wound and used it in combination with a computational model to measure the impact of paracrine communication on these responses. The experiments show that paracrine signaling by the growth factor EGF reduces the variability in the responses of cells to the ATP signal. However, this reduction is limited by the size of the paracrine communication distance. Paracrine communication distances that are too small or too large either do not provide adequate reduction in variability or result in "over-averaging". Handly et al.'s findings show that there is an optimal level of paracrine signaling during wounding that helps to coordinate the response in nearby cells without inappropriately over-averaging the signal.

necessarily detrimental to cellular function. In fact, cellular heterogeneity often plays a critical role in ensuring proper cellular response by mechanistically increasing the cellular response range to a constantly changing environment (*Altschuler and Wu, 2010*). For example, single-cell noise in NFκB dynamics creates robust population level responses to a wider range of inputs (*Hughey et al., 2014*; *Kellogg and Tay, 2015*).

Cells share information with each other via paracrine signaling, which effectively averages variable cellular responses and therefore reduces cellular variability. Overall population-level averaging decreases variability by following the statistical laws of the central limit theorem and the law of large numbers (*Piras and Selvarajoo, 2014*). Paracrine signaling averaging can decrease variability in a similar manner, but functions on a local population level. Specifically, paracrine signaling averaging functions such that the local concentration of the paracrine ligand, or the concentration of ligand a cell is exposed to, is the average ligand concentration secreted by local cells (*Figure 1A*). Indeed, the benefits from paracrine communication were previously demonstrated to increase post-paracrine cellular response fidelity (*Rand et al., 2012*; *Shalek et al., 2014*). The process of local population 'information averaging' by each cell enables increased accuracy of inherently single cell decisions such as proliferation and differentiation.

Despite promises of noise mitigation from paracrine averaging, parameters set by biological systems can limit these potential benefits. For example, population averaging due to paracrine communication may cause loss of information in a similar manner to the information loss of single-cell dynamics due to 'population average' bulk measurements (*Elowitz et al., 2002*; *Newman et al., 2006*; *Bar-Even et al., 2006*). The potential information loss due to 'over-averaging' of variable single-cell responses demonstrates a limitation to paracrine communication. Limitations to paracrine communication are also observed in post-paracrine single-cell responses that remain highly variable despite paracrine averaging. These limitations suggest an overall functional constraint to the potential benefits of paracrine communication. However, the identity and source of these limitations on paracrine communication benefits are unknown.

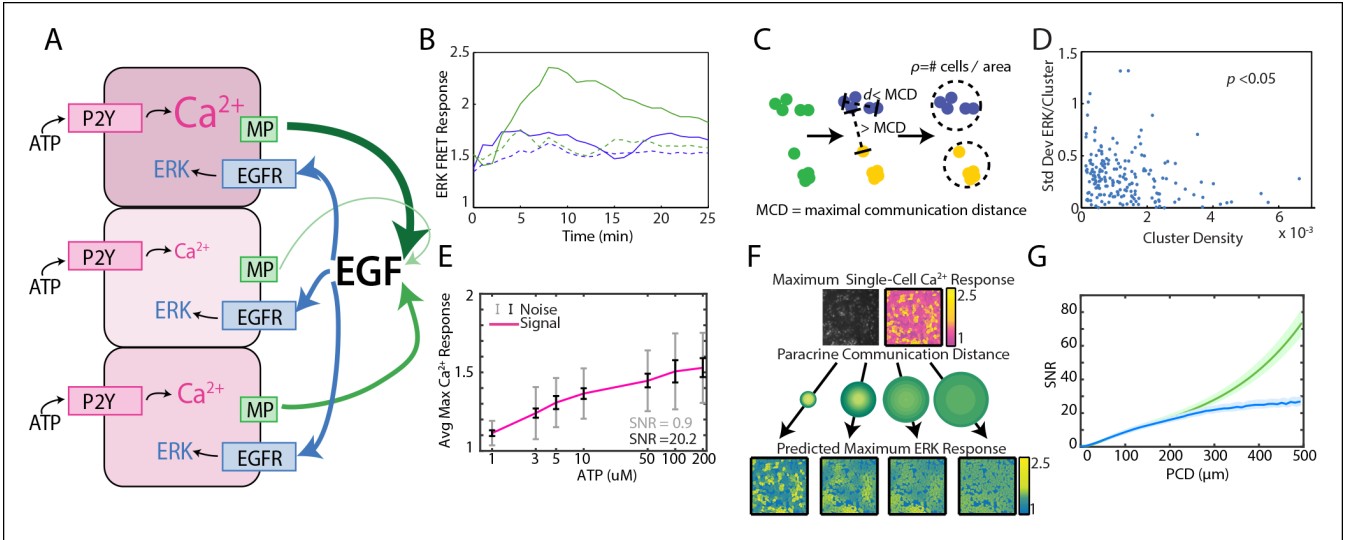

**Figure 1.** Local averaging using paracrine signaling reduces response variability in a communication distance dependent manner. (**A**) A hypothesis for local averaging based reduction of response variability using paracrine signaling for ERK activation by P2Y receptors. ATP binds to P2Y receptors to increase cytosolic $Ca^{2+}$ levels with high variability between cells despite equivalent ATP dosage per cell (pink shading). EGF release from each cell is proportional to the primary response to ATP (green arrows). Due to diffusion of EGF, the local concentration will be the average of EGF released from nearby cells and subject cells in a local neighborhood to the same level of EGF (blue arrows) to result in similar ERK activation. (**B**) ATP activates ERK in a paracrine fashion in MCF-10A cells. ERK response to 10 µM ATP addition with (green dashed) and without (green solid) 1 µM of EGFR inhibitor tryphosphitin AG1478. 0 µM ATP addition with (blue dashed) and without (blue solid) 1 µM AG1478 shown as controls. (**C**) MCF-10A cells were clustered based on their spatial proximity so that cells within a cluster were within a specific maximal communication distance (MCD) and cells in other clusters were farther than the MCD. Cluster denisty was calcuated by dividing the number of cells per cluster by the circular area inhabitated by each cluster. (**D**) Standard deviation of ERK response per cluster to 10 µM ATP. Standard deviation decreases with increasing cluster density (p-value <0.05, Pearson correlation). (**E**) Average maximum $Ca^{2+}$ response with increasing ATP dosage. The standard deviation, that is noise, of the $Ca^{2+}$ response to each ATP dosage is large when compared to the increase in average response with increasing ATP dose, that is signal (SNR = 0.9, gray error bars [standard deviation]). Noise decreases when $Ca^{2+}$ response is locally averaged with a PCD of 100 µm (SNR = 20.2, black error bars [standard deviation]). x-axis shown in log-scale. F. Single-cell maximum $Ca^{2+}$ response is locally averaged across an area specified by the PCD to produce a predicted single-cell ERK response. Variability between cells in the predicted ERK response decreases with increasing PCD. Response magnitude of $Ca^{2+}$ and ERK response indicated by pink to yellow and blue to yellow colorbars, respectively. G. The SNR of the predicted ERK response from locally averaged $Ca^{2+}$ data continually increases with increasing PCD shown for a model with rapid diffusion (green) or limited by the diffusion rates and integration time of paracrine signals (blue, Materials and methods). SNR calculated in same manner as panel E with increasing PCD. Shaded area is SEM (N = 5).

The following figure supplements are available for Figure 1:

**Figure supplement 1.** MCF-10A cells can be separated and analyzed in cell clusters when plated at low densities.

**Figure supplement 2.** Cluster standard deviation and cluster average as a function of cluster density show significant trends for ERK activation but not $Ca^{2+}$ activation.

**Figure supplement 3.** Paracrine ERK activation depends on Src prior to MMP activation.

**Figure supplement 4.** Inhibiting paracrine communication does not allow decreased cellular response variability.

**Figure supplement 5.** Mutual information and SNR both continue to increase with increasing PCD.

**Figure supplement 6.** Scaling of Paracrine Communication Distance.

**Figure supplement 7.** Required Integration time.

**Figure supplement 8.** The effect of fluid flow on paracrine communication.

**Figure supplement 9.** The effect of cellular decoding schemes on paracrine communication.

The initial paracrine signaling pathways that are activated in response to Damage Associated Molecular Patterns (DAMPs) are a good model system for investigating the influence and limits of paracrine communication on cellular response fidelity. Paracrine communication is pervasive during initial wound response. Wound healing begins as soon as the wound occurs and the initial cellular wound response provides the foundation for proper downstream healing. The initial cellular wound response relies on external environmental cues as well as programs inherent to the cell, including DAMPs as primary danger signals (*Enyedi and Niethammer, 2015*). DAMPs are released from necrotic cells and bind to extracellular receptors on surrounding cells. This binding initiates a signal in the surrounding cells to secrete a secondary set of cytokines and growth factors required to coordinate the wound healing process. Many DAMP signals, such as extracellular ATP, are transient and released in limited quantities. As a result, the initial wound response to such DAMPs shows high cellular variability and low fidelity. Despite the limited fidelity of the initial wound response, the wounded epithelium is able to establish a robust healing response. The complicated and multi-step wound healing process utilizes several paracrine communication mechanisms to share cellular information and coordinate the overall healing program.

Here we use the paracrine release of epidermal growth factor (EGF) ligands initiated by ATP binding to P2Y receptors as a model to investigate the limits of cellular information sharing through paracrine communication to mitigate biochemical noise (*Figure 1A*). We show that paracrine communication increases extracellular signal-regulated kinase (ERK) response fidelity using live single-cell quantitative fluorescent imaging of primary $Ca^{2+}$ and secondary ERK responses downstream of P2YR and EGFR, respectively. Statistical analysis of the primary response signal-to-noise ratio (SNR) demonstrates that the increase in response fidelity is limited by paracrine communication distance (PCD). To analyze this pathway in the physiological context of wound response we developed a new microfluidics device to monitor the spatial propagation of initial wound response signaling. Our results demonstrate that the interplay between the wound induced spatial signaling gradient and the cellular noise pattern produces an optimal PCD. The optimal PCD balances the benefits of decreased noise from local averaging with the cost of reduced signal of the spatial signaling gradient due to over-averaging. Empirical measurements of the PCD reveal that cellular communication occurs at a distance to maximize cellular response fidelity.

## Results

### Paracrine signaling reduces response variability

Here we establish that the paracrine activation of ERK by ATP provides a suitable system to investigate signaling response fidelity changes due to paracrine communication. Extracellular ATP binding to P2YR results in EGF family ligand release to bind EGFR and activate ERK response as monitored by ERK activity following ATP addition. In the mammary epithelial cell line MCF-10A, addition of extracellular ATP increases ERK kinase activity in an EGFR dependent manner (*Figure 1B*) similar to results reported in other in vitro epithelial models (*Yin et al., 2007*). ERK, as measured by the genetically encoded FRET sensor EKAREV (*Albeck et al., 2013*; *Komatsu et al., 2011*), increases when stimulated with ATP. Inhibiting EGFR with tryphostin AG1478 prevents ERK activation upon ATP addition showing that ERK activation depends on secreted EGF binding to EGFR (*Wetzker and Böhmer, 2003*).

With our paracrine communication system established, we next confirmed the influence of paracrine communication on cellular response variability. Under conditions of low cell density we used a spatial clustering analysis to group cells such that the distance between groups effectively constrained communication to cells within groups (*Figure 1C,D*, *Figure 1—figure supplement 1,2*). In the case that cellular coordination is beneficial, we anticipated that the ERK response within groups of higher cellular density, that is cells have increased communication ability, would have reduced response variability than cell clusters with decreased communication ability. ERK response variability within clusters decreases with increasing cluster density indicating increased intercellular communication ability. Increased intercellular communication ability is not observed in the absence of paracrine communication such as with the primary $Ca^{2+}$ response to ATP (*Figure 1D*, *Figure 1—figure supplement 2*). Furthermore, disrupting paracrine communication by partially inhibiting Src results in the loss of the observed benefit of paracrine communication in higher density clusters (*Figure 1D*,

*Figure 1—figure supplements 3,4*). Together these observations support the hypothesis that paracrine communication decreases cellular variability by increasing cellular coordination.

## Computational analysis of variability reduction resulting from paracrine information sharing

Next we developed a computational model that mimics the coordination-effects of paracrine communication (see Materials and methods for details). This computational model quantifies the overall observed benefit of paracrine coordination and predicts the potential reduction in variability. Experimental single-cell dose response data of primary $Ca^{2+}$ (prior to paracrine communication) response to ATP is used as an input to predict the secondary ERK response. We quantify cellular response fidelity by using a simple signal-to-noise analysis (SNR). In this analysis the cellular response magnitude of the input ligand (signal) is divided by the cellular response variability (noise). The signal is estimated by calculating the spread between the average cellular $Ca^{2+}$ responses from multiple ATP concentrations using multi-well dose response data. Noise is calculated from the average variability between cellular responses to a single input ligand concentration. SNR is simply the ratio of these two estimates (*Figure 1E*). To mimic the benefit of paracrine communication our computational model performs a local, spatially weighted average (convolution) of the primary $Ca^{2+}$ response to predict the variability of response post paracrine communication (ERK) (*Figure 1F*). In short, the convolution averages the signal for every cell with its associated surrounding cells by weighting the surrounding cells based on a Gaussian function parameterized with varying PCDs. The PCD represents how far the paracrine molecule travels from a single-cell to activate its associated surrounding cells. Local spatial averaging provides an upper bound of the possible benefit resulting from cellular communication in conditions where no additional noise exists in the paracrine pathway. This analysis indicates that paracrine averaging using a PCD of 100 µm increases response SNR from 0.9 to 20.2 by decreasing noise, or response variability, of the predicted ERK response (*Figure 1E*, gray/black). To investigate the limits of paracrine averaging, we repeated this analysis for multiple PCDs. Interestingly, our analysis estimates that the overall response SNR can increase up to eightyfold at PCDs of 500 µm when paracrine diffusion is not limiting, and up to twenty fivefold when diffusion of the paracrine ligand is limiting (*Figure 1G*). More sophisticated statistical measures, such as mutual information, produce similar results (*Figure 1G*, *Figure 1—figure supplement 5*). The large maximal SNR benefit suggests a potentially noise-free ERK response. However, experimental measurements of ERK response fidelity shows substantial ERK variability indicating potential factors that limit the benefit gained from paracrine communication (data not shown).

## Cellular response fidelity depends on the extent of paracrine signaling during wound response

Extracellular ATP released from necrotic cells act as DAMPs to activate healthy cells proximal to the wound (*Cordeiro and Jacinto, 2013*). Given this role, the spatial component produced by the ATP concentration gradient and the resulting cellular positional information relative to the wound may be important in the analysis of paracrine communication that occurs over hundreds of microns from the wound. Our previous SNR analysis demonstrating an increasing SNR with increasing communication distance was done based on multi-well experiment data. However, the bolus addition of ATP creates a spatially uniform ligand concentration in the well and does not represent a physiologically relevant spatial component. To examine whether ATP spatial patterns influence the paracrine communication benefit we repeated the SNR analysis using single-cell wound response data.

In order to measure the spatial wound response for epithelial cells, we first developed a convection-free, small-volume wounding device. Scratch-assays, where a monolayer of cells is mechanically wounded using a pipet tip, are traditionally used for epithelial cell wounding (*Sholley et al., 1977*). Although the scratch-assay is useful for studying cell-migration following wounding, scratch-assays lack the ability to study paracrine signaling. The large volume above the cells and convection caused by the scratch present challenges to examine paracrine signaling due to the dilution and inadvertent mixing of any paracrine molecules released from a cell into the surrounding media. To circumvent these technical issues we developed a microfluidics based wounding device (*Figure 2A,B*). Our device has two components: an air channel (black) and a cell chamber with a ~2.5 µL volume (orange). The ceiling of the cell chamber has a PDMS pillar that, when air pressure is increased in the

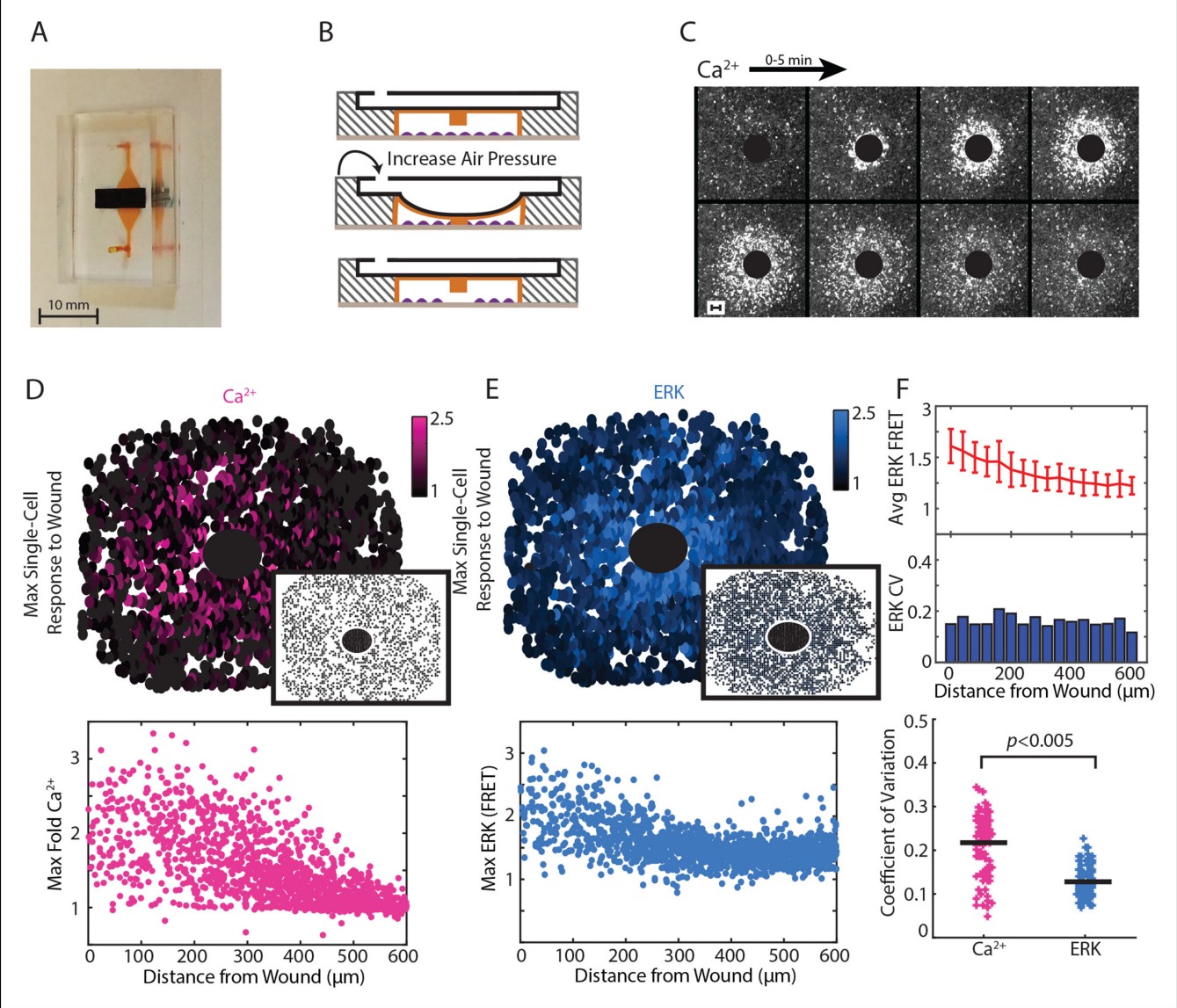

**Figure 2.** Paracrine communication reduces response variability during wounding. (**A**) Dual layer microfluidic-based wounding device with a top air channel (black) and bottom cell chamber (orange). (**B**) Schematic of wounding in the device. Cells are first loaded into the cell chamber (top). Increasing the air pressure in the air channel lowers a pillar in the ceiling of the cell chamber until cells below the pillar are mechanically crushed (middle). The pillar returns to the original height when air pressure is released (bottom). (**C**) $Ca^{2+}$ response visualized by the Fluo-4 $Ca^{2+}$ indicator dye over a period of 5 min following a 300 μm diameter wound (black circle). (**D**) Top: Maximum single-cell (dots) $Ca^{2+}$ response to a 300 μm wound. Inset shows maximum $Ca^{2+}$ response to 300 μm wound in the presence of the ATP scavenger apyrase. Bottom: Maximum single-cell (dots) of $Ca^{2+}$ response to wound according to distance from the wound. (**E**) Same as D but for maximum ERK response. (**F**) Top: Cells are binned according to distance from the wound (*Figure 3A*) and the average and standard deviation (error bars) are found for each bin. Middle: Coefficient of variation (CV) calcuated by dividing the standard deviation of each bin by the mean of that bin. Bottom: $Ca^{2+}$ has higher variability than ERK response for the wound according to the CV of every bin for all wounds (Black bar = average CV, p-value by t-test).

The following figure supplements are available for Figure 2:

**Figure supplement 1.** Microfluidic wounding device characterization demonstrates cell viability, isotropic wounding, wounding control, and reproducibility.

upper air channel, lowers down on to the cells, thereby wounding the cells in the cell chamber in a highly controlled and reproducible manner (*Figure 2B,C*, *Figure 2—figure supplement 1*, *Video 1*).

We used our wounding device to monitor $Ca^{2+}$ and ERK response to a 300 μm diameter wound using a stable, dual reporter MCF-10A cell line expressing the genetically encoded $Ca^{2+}$ indicator RGECO (*Akerboom et al., 2013*; *Zhao et al., 2011*) and the EKAREV FRET reporter for ERK (*Albeck et al., 2013*; *Komatsu et al., 2011*) (*Figure 2D,E*; *Video 2*). We verified the key role of ATP in initial wound response by wounding in the presence of apyrase, an enzyme that rapidly hydrolyzes ATP. Wounding in the presence of apyrase inhibits both $Ca^{2+}$ and ERK response (*Figure 2D,E*, insets). From each wound we quantified single-cell time traces for over 3000 cells. Notably, the maximum activity per cell shows a larger response in cells closer to the wound compared to cells farther away from the wound for both $Ca^{2+}$ and ERK. These response gradients demonstrate the importance of the cellular position to determine the cellular response, or positional information (*Figure 2D,E*). We used coefficient of variation (CV) to measure the variability of the post-paracrine ERK response and the pre-paracrine $Ca^{2+}$ response in the wound (*Figure 2F*). Indeed, the CV for $Ca^{2+}$ wound response shows statistically higher variability than ERK wound response indicating that paracrine communication reduces response variability during initial wound response.

We adapted the computational SNR analysis to wound response data to determine the influence of spatial patterns on response fidelity. As opposed to the dose-response data, the wound response data uses the distance of each cell from the wound as the input rather than the concentration of activating ligand (*Figure 3A*). Similar to the dose-response data, noise is estimated by averaging the cellular response variability over all distances. The variability between the average response magnitude of each distance constituted the signal (*Figure 3B*). Other statistical measures of response fidelity such as mutual information were also adapted for the wound context (*Figure 3—figure supplement 1*).

The maximum primary $Ca^{2+}$ response shows highly variable cellular response when plotted according to distance (*Figure 3C*, pink). This variability complicates the ability for a cell to distinguish its respective position to the wound based on its response We again mimicked paracrine communication to predict the post-paracrine ERK response by locally averaging the single-cell $Ca^{2+}$ wound response using a Gaussian kernel (*Figure 1F*). Locally averaging the cellular $Ca^{2+}$ response creates a smoother predicted ERK response pattern versus distance from the wound (*Figure 3C*, gray). However, the reduction in variability also decreases the overall response pattern trend. Locally averaging the $Ca^{2+}$ signal using increasing PCDs decreases the magnitude of change of the average predicted ERK response between cells closest to the wound and farthest from the wound (*Figure 3D*). In other words, the response gradient becomes less obvious when cells are averaged over larger distances. Therefore, although the increase in PCD decreases response noise, the corresponding decrease in signal demonstrates the limit of PCD on the SNR benefit (*Figure 3E*). The difference in rates at which the signal and noise decrease results in a maximum SNR at a PCD of 91.0+/-6.3 μm (SEM, N = 5) (*Figure 3F*). This peak corresponds to a PCD where the amount of noise is decreased to the lowest amount possible without reducing the response gradient due to 'over-averaging'. The

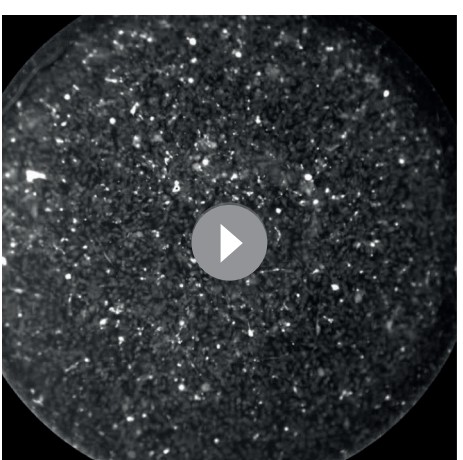

**Video 1.** Isotropic $Ca^{2+}$ response to wounding. $Ca^{2+}$ response to a 300 μm wound, indicated by the Fluo-4 $Ca^{2+}$ sensor. Upon wounding, $Ca^{2+}$ response propagates isotropically from the wound. Movie time lapse is 5 min.

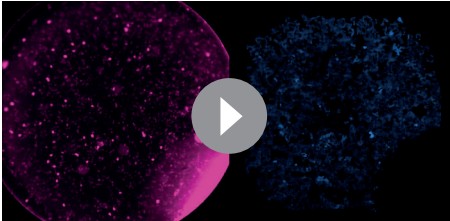

**Video 2.** $Ca^{2+}$ and ERK dual wounding. $Ca^{2+}$ and ERK are measured simultaneously using the fluoresecent reporter RGECO for $Ca^{2+}$ (pink) and the ERK FRET sensor EKAREV (cyan). $Ca^{2+}$ response is completed within 5 min whereas ERK response takes approximately 30 min.

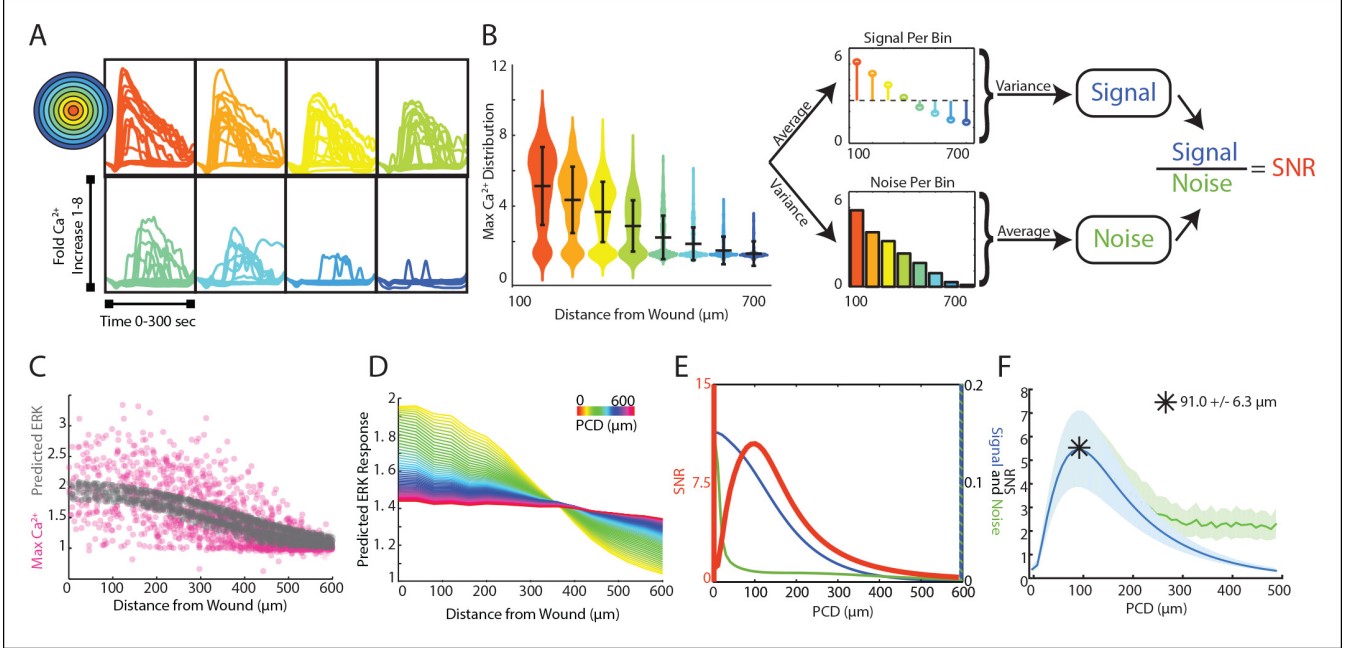

**Figure 3.** Signal to noise analysis of initial wound response shows limits to paracrine communication. (**A**) Representative single-cell time traces of $Ca^{2+}$ response to wounding, grouped according to distance from the wound (concentric circle colors). (**B**) SNR calculation method for $Ca^{2+}$ response adpated to the wound. Horizontal bars represent bin average and error bars represent bin variance. (**C**) Maximum single-cell (dots) $Ca^{2+}$ response to wound with respect to distance away from the wound (pink). Predicted ERK cellular response after paracrine communication as determined by local averaging (gray). Local averaging done in same manner as *Figure 1F*. (**D**) Predicted ERK response according to distance from the wound using PCDs of 0 to 600 µm (colorbar). Predicted ERK response determined through local averaging using increasing PCDs results in decreased response magnitude over space. (**E**) Signal (blue), Noise (green) and SNR (orange) as function of PCD of locally averaged $Ca^{2+}$ response trends in panel D. (**F**) SNR analysis of locally averaged $Ca^{2+}$ response to a wound with increasing PCD shows a maximum SNR at PCD of 91.0 µm+/-6.3 µm indicated by the asterick (blue, SEM indicated by shaded region, N = 5). The maximum SNR for conditions controlled for biologically relevant integration times show the same maximum SNR (green, Materials and methods).

The following figure supplements are available for Figure 3:

**Figure supplement 1.** Mutual information analysis of locally averaged $Ca^{2+}$ response to wounding shows similar peak to SNR analysis.

predicted PCD with maximal benefit did not change when we expand the model to consider limitations due to diffusion (*Figure 3F* green curve, Materials and methods). Similar analysis using mutual information statistics shows a similar PCD with the maximal mutual information at the distance that showed maximal SNR (*Figure 3E,F*, *Figure 3—figure supplement 1B*). This analysis shows that the benefits from paracrine communication depend on how far a paracrine molecule travels which, in this specific case, has a maximal benefit at ~100 µm, or approximately three cell diameters.

## Direct measurement of Paracrine Communication Distance

We next empirically measured the PCD in our experimental system to compare to the PCD predicted to maximize the SNR in the wound context. To measure the PCD of ERK activation we first established a co-culture system that allows us to separate the effects of autocrine and paracrine signaling. Our assay utilizes a synthetic GPCR: Designer Receptors Exclusively Activated by Designer Drugs (DREADD). The Gq human muscarinic derived GPCR DREADD is activated by a synthetic small molecule, clozapine-N-oxide (CNO), that has no known endogenous receptors (*Armbruster et al., 2007*). In addition, DREADD activates the Gq pathway similar to purinergic ATP receptors (*Dong et al., 2010*). Using a co-culture of DREADD expressing (activated by CNO) and non-expressing cells (not activated by CNO), we can determine which cells release EGF (DREADD expressing-red) and which cells accept EGF (non-expressing-gray) (*Figure 4A*). Using a synthetic system allows us to directly measure the average communication distance of EGF. CNO addition selectively activates $Ca^{2+}$ response in DREADD expressing cells while the surrounding non-expressing cells show

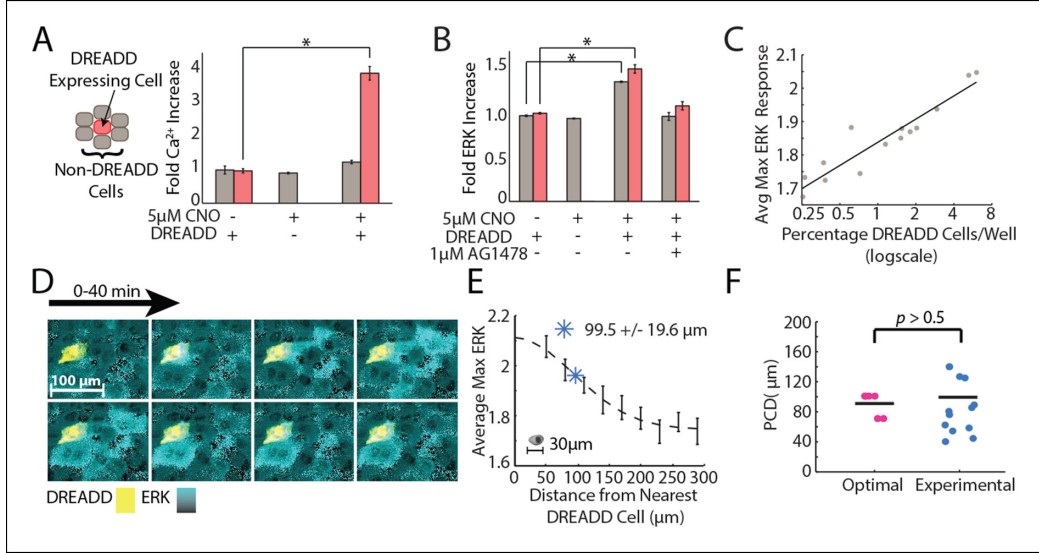

**Figure 4.** Empirical PCD measurement using DREADD synthetic GPCR show that cells use an optimal level of paracrine communication to maximize cellular response fidelity. (**A**) The addition of 5 μM CNO to a co-culture of DREADD expressing (red) and non-expressing (gray) MCF-10A cells shows increased fold $Ca^{2+}$ response in DREADD expressing cells but not non-expressing cells (SEM indicated by error bars, N = 3; *p-value<0.005, t-test). (**B**) Fold ERK increase in DREADD co-culture assay. Both DREADD and non-expressing cells show significant ERK increase when both DREADD cells and 5 μM CNO are present. ERK activation inhibited by 1 μM AG1478 (SEM indicated by error bars, N = 3; *p-value <0.005, t-test). (**C**) The average maximum ERK response of non-expressing cells in a given well increases linearly with an increasing percentage of DREADD cells per well. (**D**) Representative images from a timelapse experiment showing ERK activation (cyan) in non-expressing cells surrounding a single activated DREADD cell (yellow) over a 40 min time period. ERK activation level indicated by black to cyan colorbar. (**E**) Average maximum ERK activation in non-expressing cells surrounding single DREADD cell according to distance from the DREADD cell. PCD was calcuated as the spread, or sigma, of the fitted Gaussian curve (dashed line) and measured to be 99.5 μm+/-19.6 μm (blue *) (SEM indicated by error bars, N = 12 DREADD cells). Scale bar represents average length of a single-cell. F. Comparison between calculated optimal PCD per wound (pink, *Figure 3F*) and experimentally measured PCD found using the DREADD co-culture assay per DREADD cell (blue, *Figure 4E*) (p-value>0.5, t-test). Horizontal bars represent average.

no response indicating a lack of paracrine activation of $Ca^{2+}$ response in cells (*Figure 4A*). Although some systems show that $Ca^{2+}$ response can propagate from cell-to-cell through gap junctions (*Ross 2012*), this does not appear to be the case in MCF-10A cells as non-expressing cells showed no cytosolic $Ca^{2+}$ increase upon activation of DREADD cells. Alternatively, ERK response was found in both DREADD expressing and the surrounding non-expressing cells upon CNO addition but was inhibited in both cell types in the presence of the EGFR inhibitor tryphosphitin AG1478, confirming paracrine activation of ERK in the DREADD system (*Figure 4B*). The $Ca^{2+}$ and ERK responses in the DREADD system suggest that local averaging takes place only at the EGF level between $Ca^{2+}$ and ERK response. Additionally, increasing the ratio of DREADD cells to non-expressing cells shows an increasing ERK response magnitude in non-expressing cells, further supporting that paracrine communication locally activates ERK (*Figure 4C*).

We measured the PCD by monitoring local paracrine ERK activation with our DREADD co-culture assay. We co-cultured DREADD cells at a low concentration compared to non-expressing cells to ensure that neighboring non-expressing cells were activated only by a single DREADD cell. We then analyzed the ERK response of ~1500 non-expressing cells neighboring a DREADD cell. (*Figure 4D, E*). Local ERK activation of non-expressing cells surrounding a DREADD cell show decreasing response with increasing distance from the DREADD cell. ERK response as a function of distance follows a Gaussian fit, consistent with how the concentration of diffusing molecules, like in paracrine signaling, changes over distance (*Figure 4E*) (*Berg, 1993*). The PCD was determined by calculating the spread, or sigma, of this Gaussian curve. According to our fit, the paracrine activation of ERK has

a communication distance of 99.5+/-19.6 µm (SEM, N = 12 DREADD cells). This empirically measured value is statistically similar to the predicted communication distance value that maximizes the SNR of wound response (*Figure 4F*). In other words, the cellular communication distance is tuned to maximize the overall response fidelity during wound response signaling.

## Discussion

Multicellular organisms utilize cellular diversity for specialization and division of labor. However, the variability between cells can be detrimental due to the potential loss of response fidelity (*Uda et al., 2013*; *Hansen and O'Shea, 2015*; *Voliotis et al., 2014*; *Cheong et al., 2011*; *Selimkhanov et al., 2014*). Paracrine communication can serve to share information between cells to regulate cellular variability. In this study we analyzed the benefits and limitations of paracrine communication based information sharing between cells as a mechanism to control cellular response variability.

We analyzed the limits of paracrine communication on cellular response fidelity in two cases. First we analyzed the response to a spatially uniform ligand. Our analysis reveals that, under these conditions, the magnitude of single-cell response fidelity increases as a function of the PCD with no upper bound. In order for cells to facilitate larger PCDs, cells would need to synthesize larger amounts of paracrine signaling molecules or utilize fast diffusing paracrine ligands like $H_2O_2$ (*Enyedi and Niethammer, 2015*). The increased energy required to synthesize the additional molecules is likely to be minor in comparison to the overall energetic demand of a cell. Therefore cells could potentially take advantage of large PCDs to substantially mitigate biochemical noise. However, a spatially uniform input, while common in cell culture experiments, is likely an inadequate representation of physiological conditions. In the second case we analyzed the response of cells to spatially defined inputs in the form of a mechanical epithelial wound. We analyzed the cellular response to extracellular ATP gradients, a damage associated molecule, following a controlled wounding of an epithelial monolayer in vitro. Similar to developmental systems, an extracellular input ligand conveys positional information in wound response (*Dubuis et al., 2013*; *Sonnemann and Bement, 2011*). Analysis of the paracrine communication benefit in our novel quantitative wound response assay with defined spatial perturbations demonstrates that paracrine communication increased cellular response fidelity, but with limitations. Unlike the spatially uniform ligand in the first case, the magnitude of the response fidelity benefit varies with increasing PCD. The maximal increase of cellular response fidelity occurrs at a PCD of ~100 µm, or approximately 3 cell diameters. In vivo work measuring ERK propagation using the same EKAREV FRET sensor also showed propagation extending ~100 µm (*Hiratsuka et al., 2015*). Our results demonstrate that the paracrine information sharing benefit depends on the input ligand spatial scale, or PCD. Furthermore, empirical measurements of paracrine communication match the physiologically relevant spatial wound response maximum communication distance.

The process of wound healing is a complex multi-stage program that coordinates the action of multiple cell types over multiple timescales, from minutes to weeks, to address an acute need. The initial steps of wound healing programs propagate information concerning the wound in a manner that is appropriate to the magnitude of damage. Both inflammatory and fibrotic processes, critical steps in wound response signaling, are damaging when they go awry. Therefore, the initial cellular responses and the establishment of signaling gradients are key steps in wound healing. The mechanistic details underlying how tissues robustly match the wound response magnitude to the extent of wound-induced damage remain unknown. Our results demonstrate that intercellular communication during the initial wound response is optimized to increase overall response fidelity and provides the initial evidence that matching the wound response to wound damage is a critical aspect to wound healing programs. Future work is needed to further investigate tissue level response fidelity during wound healing programs.

Paracrine communication increases the fidelity of response at the single-cell level by mitigating biological noise at the single-cell level. Each cell integrates information from its local neighborhood to increase its individual response fidelity. Local averaging at the cellular level is a distinct mechanism compared to the benefit of global averaging at the population level. Without any paracrine communication, the reliability of the average of a cell population response can only increase with the size of the population. This is a consequence of the central limit theorem where the uncertainty of a sample average decreases with sample size. However, this increase in reliability is only true for the

population average and not for individual cells in the population. Therefore, in cases where the biologically significant output is the collective action of the population, for example the secretion of a cytokine, intercellular information sharing is not required. However, when biologically significant output requires single-cell action, local information sharing via paracrine communication increases cellular response fidelity. Therefore, whether paracrine communication is required remains context dependent. It is possible that paracrine information sharing is more prevalent in signaling networks that support individual cellular decisions and less prevalent in cases where biologically meaningful outcomes result from population averages.

In cases where paracrine information sharing is used as a method to mitigate biological noise, the breakdown of this system could be detrimental. In vivo studies of ERK response in mammary tumor cells using the same EKAREV FRET sensor utilized here show highly variable ERK response that may lead to the survival and propagation of cancer cells (*Kumagai et al., 2014*). Although the cause for this heterogeneity is unknown, one possible mechanism may be the breakdown of paracrine communication between cells similar to how our partial inhibition of paracrine communication showed no decrease in ERK variability (*Figure 1—figure supplement 4*).

The abundance of paracrine communication in mammals, that is the activation of a receptor by a ligand synthesized by another cell, demonstrates the heavy utilization of intercellular communication (*Ben-Shlomo et al., 2003*; *2007*). Paracrine averaging demonstrates how intercellular communication enables cellular collective decision making where the 'wisdom of the crowd' is greater than the individual cell. Theoretical and empirical work in humans and animal collectives has shown that the benefit of collective decision making depends on the size of the group; big crowds are not always better than small crowds (*Sasaki et al., 2013*; *Kao et al., 2014*; *Hoare et al., 2004*; *Sueur et al., 2011*). Therefore, it is likely that the extent of secretion of each paracrine ligand is adjusted to the level of cellular information sharing to ensure an effective collective decision.

The optimal PCD we identified is not universal. Rather, the optimal distance depends on the specific shape of the spatial pattern of the initial activating ligand and the noise pattern of the primary response. Additionally, propagation patterns of the same activating ligand can depend on the physiological signaling context as demonstrated by differences found during in vivo ERK propagations under wound and normal conditions (*Hiratsuka et al., 2015*). The effective PCD can be regulated at the cellular level by several possible factors to optimize the benefit of paracrine communication to the specific noise and spatial patterns characteristic to each signaling system (*Batsilas et al., 2003*; *Muratov and Shvartsman, 2003a*; *2003b*). PCD also depends on the effective diffusion coefficient of the secreted molecule, transmitted signal strength (e.g. number of secreted molecules), and receiver cell sensitivity (e.g. receptor $K_d$). The diffusion coefficients of paracrine signaling molecules can vary by two orders of magnitude, let alone differences in signal strength and receptor sensitivity in individual paracrine signaling pathways (*Kreuz et al., 1965*; *Gregor et al., 2007*). Fine-tuning each of these factors provides a possible mechanism for cells to regulate the PCD and thereby the extent a cell locally communicates. The ability to specifically tune PCD raises the possibility that evolutionary pressures can tune paracrine communication to provide the optimal benefit in many other paracrine communication systems.

## Materials and methods

### Ca$^{2+}$ and ERK measurements in MCF-10A cells

MCF-10A cells were cultured following established protocols (*Debnath et al., 2003*). Before plating cells, each surface was first treated with a collagen (Life Technologies, Carlsbad, CA), BSA (New England Biolabs), and fibronectin (Sigma-Aldrich) solution in order for cells to completely adhere, according to established methods. In order to maintain a viable environment, cells were imaged at 32°C and 5% $CO_2$. All EGF (PeproTech) titrations and DREADD experiments were conducted in 96-well plates using extracellular hepes buffer (ECB) to reduce background fluorescence (5 mM KCl, 125 mM NaCl, 20 mM Hepes, 1.5 mM $MgCl_2$, and 1.5 mM $CaCl_2$, pH 7.4). All imaging for wounding was done in MCF-10A assay media (*Debnath et al., 2003*).

### ERK and Ca²⁺ activation by DREADD

Cells were plated at a density of 2,000,000 cells/100mm plate and allowed to adhere overnight. Cells were transfected with the Gq-coupled DREADD HA-tagged hM3D with an mCherry tag using a 3:1 ratio of FuGene HD (Promega) to DNA and allowed to incubate overnight (*Dong et al., 2010*). In order to measure the paracrine signal from a single-cell, non-transfected cells were mixed with DREADD-transfected cells at ratios of 1:0, 1:1, 1:2, 1:5, 1:7, and 0:1 (non-transfected:DREADD) and plated in 96-well plates at a density of 30,000 cells/well. The following day, cells were loaded with 1 µM Hoechst dye for nuclear imaging for 30 min for cell segmentation purposes. 5 µM clozapine-N-oxide (CNO) (Enzo Life Sciences) was added to each well to specifically activate DREADD cells. ERK activation was monitored using the EKAREV FRET reporter (*Albeck et al., 2013*; *Komatsu et al., 2011*) and Ca²⁺ activation was monitored using the Ca²⁺ indicator dye Fluo-4 using the published protocol (Invitrogen).

### Cell clustering assay and analysis

In order to measure the standard deviation of Ca²⁺ and ERK activity within a small group of cells, MCF-10A cells were plated at densities of 1000, 2000, and 3000 cells per well in a 96-well plate, taking advantage of the natural tendency for MCF-10A cells to cluster together. Cells were stimulated with 10 µM or 100 µM ATP and imaged for 5 min every 3 s (Ca²⁺) or 30 min every minute (ERK).

Standard deviation and average expression of Ca²⁺ and ERK were analyzed by grouping cells in to clusters based on the distances between cells and clusters (*Figure 1—figure supplement 1*). Following the cluster analysis, the average and standard deviation of Ca²⁺ and ERK activation were calculated for each cluster. ERK activation was measured using the ERK FRET reporter (*Albeck et al., 2013*; *Komatsu et al., 2011*) and Ca²⁺ activation was monitored using the genetically encoded sensor R-GECO (*Zhao et al., 2011*).

### Wounding device design, fabrication, and wounding assay

Master molds for the microfluidics based wounding device were created using silicon wafers and layer-by-layer photolithography using established methods (*Ferry et al., 2011*). A separate mold for both the air layer and cell layer were made using negative photoresists and masks. Chips were made by pouring uncured polydimethylsiloxane (PDMS) onto each mold, allowing the PDMS to harden, and bonding the layers together and subsequently to a glass slide. Cells were loaded into the devices through the inlet port using a 20G needle. During wounding the outlet port was plugged using tape and the inlet port held a reservoir of media to prevent evaporation in the chamber. Wounding was accomplished by increasing the air pressure in the top layer of the device until the pillar made contact with the bottom of the device after which the air pressure was released to raise the pillar back up. Cells were loaded in to the wounding device at a density of 15,000,000 cells/mL using a 20G needle. Following trypsinization and resuspension, cells were put on ice to prevent aggregation. Two o-rings were attached to the device surrounding both the inlet and outlet ports for media reservoirs. Each o-ring was attached using a thin film of vacuum grease. Wounding devices were kept in an empty pipet box filled with water to prevent media evaporation. Cells were allowed to adhere for 18–24 hr before wounding.

### Imaging and image analysis

Imaging was accomplished using a Nikon Plan Apo λ 10X/0.45objective with a 0.7x demagnifier and Nikon Eclipse Ti microscope with a sCMOS Zyla camera. All imaging was accomplished using custom automated software written using MATLAB and Micro-Manager (*Edelstein et al., 2010*). Image analysis was accomplished using a custom MATLAB code published previously (*Selimkhanov et al., 2014*) and is available through GitHub repository https://github.com/rwollman/CellSegmentation.git.

### Model for paracrine communication based on local isotropic diffusion

The paracrine ligand concentration (P) for a cell at position (x,y) observed by (P[x,y]) is the local average of the concentration of ligand released by cells in the local neighborhood (*Figure 1A*). We modeled this paracrine ligand local average using a convolution of two functions: $S(x,y)$ that represents

the amount of ligand secreted by each cell and $D(x,y)$ that represents the expected diffusion of the paracrine ligand during the timescale of paracrine signal integration:

$$P(x,y) = \iint du\, dv\, S\,(x,y)\, D\,(x-u, y-v) \tag{1}$$

The function S(x,y) was estimated using experimental cellular Ca$^{2+}$ response data according to:

$$S(x,y) = \begin{cases} \max\left(Ca_i^{2+}(t)\right) & x,y \;\in\; cell\;\; i \\ 0 & x,y \;\in\; background \end{cases} \tag{2}$$

The function D(x,y) was approximated to follow Gaussian weights with a length-scale we named the Paracrine Communication Distance (PCD):

$$D(x,y) = \frac{1}{PCD\sqrt{2\pi}} e^{\frac{-(x^2+y^2)}{PCD^2}} \tag{3}$$

Detailed analysis of how PCD depends on diffusion, number of secreted paracrine molecules, sensitivity of detection, physiological levels of fluid flow (*Polacheck et al., 2011*), and cellular decoding of time varying paracrine signal are presented in Materials and methods. In cases where biologically relevant integrations times may influence the predicted paracrine communication response, the PCD did not exceed the approximate distance EGF could travel before the first ERK response (*Figure 1G*, *Figure 3F*, Materials and methods). Based on single-cell ERK data to ATP stimulation, this time was found to be ~5 min which resulted in a maximum PCD of ~300 µm (data not shown) based on EGF diffusion coefficient of 50 µm$^2$/s.

## Signal to noise analysis

Signal to Noise ratio analysis on Ca$^{2+}$ response to ATP titration data was estimated as was done previously (*Selimkhanov, 2014*). Briefly, the signal S was calculated using:

$$S = \operatorname*{var}_{bins}\left(\operatorname*{avg}_{cells}\left(\max_t\left(Ca^{2+}(t)\right)\right)\right) \tag{4}$$

The noise N was calculated by:

$$N = \operatorname*{avg}_{bins}\left(\operatorname*{var}_{cells}\left(\max_t\left(Ca^{2+}(t)\right)\right)\right) \tag{5}$$

Where Ca$^{2+}$ (t) is the temporal time series of Ca$^{2+}$ response measured experimentally. Cells are separated into bins according to either different dosages of ATP added to multiple wells (*Figure 1*) or different distances from the wound source (*Figure 3*). SNR was then simply: SNR = S/N.

## Analysis of the effects of diffusion, secretion, and integration time on paracrine communication

In this section we analyze how the Paracrine Communication Distance (PCD), the characteristic length-scale of paracrine communication, depends on factors related to the paracrine signal. Specifically we look into how the PCD depends on the diffusion coefficient D, the number of molecules released from a cell $N_r$, the number of molecules needed for detection $N_d$, and the total integration time T.

To understand how PCD depends on the factors mentioned above, we considered the diffusion of a paracrine ligand from a single cell to its surrounding neighbors. We considered a 2D-like geometry where cylindrical cells, each of height $h_c$ and radius $\rho$, grow in a chamber of total $h_f$ height. We simplify the below analysis by approximating the cell monolayer geometry to a series of 'cell cylinders'. The key results of the scaling of PCD and required integration time are similar for other comparable geometries (data not shown). Under these conditions one could write the analytical solution of the diffusion equations:

$$C(r,t) = \frac{N_r}{h_f \cdot 4D\pi t} e^{-\frac{r^2}{4Dt}} \tag{6}$$

Where $C(r, t)$ is the concentration of paracrine ligand for distance $r$ and time $t$. For a neighboring cell to respond to this paracrine signal, a critical number of molecules $N_d$ needs to reach the volume surrounding the cell. We assume that a cell 'senses' a volume comparable to the volume of a cell itself. For a cylindrical cell of area $\pi\rho^2$ and height $h_c$ the critical concentration required for cellular response will be:

$$C_{detect} = \frac{N_d}{h_c \pi \rho^2} \tag{7}$$

This is simply the required number of molecules divided by the cell volume. Combining *equations 6,7* we can solve for the distance and time of where the critical concentration will be reached. Solving for distance we get that

$$rdetect = 2\sqrt{Dt \ln\left(\frac{\rho^2 h_c N_r}{4DtN_d h_f}\right)} \tag{8}$$

The concentration of the paracrine ligand is diluted as it diffuses from the source. Therefore, there is a point in space which is the maximal distance from the source that the critical detection concentration $C_{detect}$ will be reached at some point in time. Distances that are greater than the critical distance will only experience concentrations lower than the critical detection concentration $C_{detect}$. The existence of such maximum can also be seen by the non-monotonous dependency of $r_{detect}$ on $t$ in *equation 8*. To find the maximal distance we can simply find the maximum of 1.3 in respect to t. Doing so we get that:

$$PCD = e^{-\frac{1}{2}} \rho \frac{\sqrt{h_c}\sqrt{N_r}}{\sqrt{h_f}\sqrt{N_d}} \tag{9}$$

We can simplify the analysis by the introduction of two dimensionless variables: 1) $S = \frac{N_r}{N_d}$ represents the strength of the signal and is defined as the ratio of released molecules $N_r$ and the number of molecules needed to detect the signal $N_d$. 2) $\eta = \frac{h_c}{h_f}$ represents the fraction of the height of the flow chamber that cells occupy. When we substitute the new variables into *Equation 1,4* we get that:

$$PCD = \frac{1}{2} e^{-\frac{1}{2}} \rho \sqrt{\eta S} \tag{10}$$

Interestingly this shows that the value of PCD does not depend on the diffusion coefficient. Rather, PCD scales as a function of the square root of the strength of the signal S with a multiplicative constant that depends on the specific cell geometry. PCD also depends on cell geometry with the cell radius $\rho$ and the relative height of a cell in the effective environment $\eta$. *Figure 1—figure supplement 6* shows *equation 10* graphically.

While the analysis above shows that the diffusion coefficient has no influence on the overall PCD, the time required to reach this maximal distance has important biological implications. 'Paracrine averaging' requires cells to integrate the signal. However, the time required for signal integration must be biologically feasible given the cellular response time and diffusion coefficient.

From *equation 8* we can identify the time by which the PCD is maximal to be:

$$T_{int} = \frac{\rho^2 \eta S}{4eD} \tag{11}$$

The integration time grows linearly with signal strength S. This is because the PCD itself scales as a square root of S and the diffusion time grows with the square of the distance. The integration time decreases with increasing diffusion coefficient as expected. *Figure 1—figure supplement 7* shows the scaling of the integration time with the diffusion coefficient for a few PCD values.

For diffusion coefficient values of ~10–100 $\mu m^2$/s and a PCD of 100 $\mu m$ (similar to the distance measured in *Figure 3*) integration times ranged between 0.5 to 5 min. Given that ERK activation is observed only after 5 min post-activation, the required integration time does not pose an issue. However, larger PCDs will require higher diffusion coefficients to allow proper integration of the paracrine signal. Interestingly, $H_2O_2$, another key paracrine signaling molecule critical to initial

wound response signaling, has a diffusion coefficient of ~2000 μm²/s. A larger diffusion coefficient could allow for a much longer PCD with reasonable biological integration times.

## Analysis of the effect of fluid flow on paracrine communication

All the analysis above assumed static conditions, that is no fluid mixing or advection of any kind. In this section we analyze the degree to which the principles of paracrine communication are applicable in non-static conditions.

Non-static fluid conditions potentially have two effects on mass transport. 1) Non-static fluid conditions can create mixing due to turbulence and 2) Laminar advection can transport secreted molecules away from the secretion source. Since the extracellular environment is characterized by a low Reynold's number there is effectively no turbulent mixing in biologically relevant parameters.

To analyze the relative contribution of advection and diffusive transport we utilize a dimensionless number, the Péclet number ($P$), that represents the ratio between the contribution of advection and diffusion:

$$P = \frac{vL}{D} \qquad (12)$$

Where $v$ is the interstitial flow rate, D is the diffusion coefficient and L is the characteristic length scale. In our case, the characteristic length scale is the PCD, which depends on the signal strength as described above (*equation 10* and *Figure 1—figure supplement 6*). Therefore, the P number can be expressed as a function of the signal strength S and diffusion coefficient:

$$P = \frac{\sqrt{S\eta}\rho v}{D\sqrt{e}} \qquad (13)$$

Graphical representation of this expression is shown in *Figure 1—figure supplement 8* where the map of D and S is color coded by the Péclet number with three highlighted regions: A red region where flow will dominate, a cyan region where diffusion will dominate, and the region in between where both advection and diffusion contribute to paracrine communication.

To gain further insight into the relative contribution of advection and diffusion we looked at the distance molecules will travel via advection for a specific signal strength (S = 1000). As can be seen in *Figure 1—figure supplement 8b*, for diffusion coefficients of small protein ligands advection will contribute minimally.

When considering positional accuracy of cellular response, an important consideration is that advection can potentially 'shift' the effects of paracrine signaling downstream of the flow. Even if the shift is characterized by low Péclet number, advection can interfere with positional information accuracy (as analyzed in *Figure 3*). To estimate the potentially degrading effects of flow we calculate the expected level of positional accuracy error induced by flow. We estimate that the wound induced signaling gradient (*Figure 3C*) to be >500 μm. Therefore the effect on positional accuracy will be minimal (<10%) at advection distances up to 50 μm, or for a PCD of 100 μm, a Péclet number up to 0.5. The isocline of a Péclet number of 0.5 is shown in *Figure 1—figure supplement 8a* as a dotted black line. This shows that for paracrine ligands with a diffusion coefficient >40 μm²/sec, advection will have little effect on positional accuracy of initial wound response signaling.

## Analysis of the effect of cellular decoding schemes on paracrine communication

The analysis in the previous two sections assumes that the concentration of the paracrine ligand decreases over increasing distance from the source of secretion according to a Gaussian fit where the diffusion length-scale represents the PCD. The cellular response to a paracrine ligand depends on cellular decoding of the temporal paracrine concentration profile a cell observes. As both the temporal profile of the secreted paracrine molecule and the temporal cellular decoding are unknown, we consider the simple assumption of a Gaussian profile reasonable. To quantitatively test this assumption we compared the Gaussian profile to an alternative model that could be addressed analytically. In the alternative model, we assume that all paracrine molecules are released at T = 0 and that cellular decoding of the paracrine signal is simple temporal averaging. Under these assumptions one can write an expression of the temporal average of the paracrine concentration at a distance r from the source as:

$$C_{avg}(r) = \int_0^{t_0} C(r,t) = \int_0^{t_0} \frac{N_r}{h_f \cdot 4D\pi t} e^{-\frac{r^2}{4Dt}} = -\frac{N_r}{4D\pi h_f \, t_0} \mathrm{ei}\left(-\frac{r^2}{4Dt_0}\right). \tag{14}$$

Where all symbols follow *equation 6* and ei represent the exponential integral: $\mathrm{ei}(x) = \int_{-\infty}^{x} \frac{e^t}{t} dt$

Comparison of the two models can be seen in *Figure 1—figure supplement 9*. Overall, the two models generate very similar Paracrine Averaging Weights (the effect of cellular decoding of paracrine signal). There is a small discrepancy between the two models at very low distances (<50 μm). However, this discrepancy is most likely a result of the assumption in the alternative model that all the paracrine molecules are released at once. Under the more realistic assumption where paracrine molecule release duration is not much smaller than the time to diffuse 50 μm (12.5 s at D = 50 μm$^2$/s) we anticipate that the similarity between these two profiles will further increase.

## Acknowledgment

We thank Jeff Hasty, Ryan Johnson, and Megan Dueck from the San Diego Center for Systems Biology Cell Dynamics core for their technical assistance. The work was supported by GM111404 & EY024960 (RW), and a training grant (GM007240) for LNH.

## Additional information

### Funding

| Funder | Grant reference number | Author |
| --- | --- | --- |
| National Institute of General Medical Sciences | GM111404 | Roy Wollman |
| National Institute of General Medical Sciences | Training grant (GM007240) | L Naomi Handly |
| National Eye Institute | EY024960 | Roy Wollman |

The funders had no role in study design, data collection and interpretation, or the decision to submit the work for publication.

### Author contributions

LNH, Acquisition of data, Analysis and interpretation of data, Drafting or revising the article; AP, Contributed unpublished essential data or reagents; RW, Conception and design, Analysis and interpretation of data, Drafting or revising the article

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
