## [Decision Letter]

Thank you for submitting your work entitled "Paracrine Communication Maximizes Cellular Response Fidelity in Wound Signaling" for peer review at *eLife*. Your submission has been favorably evaluated by Naama Barkai (Senior Editor), a Reviewing Editor, and two reviewers.

The reviewers have discussed the reviews with one another and the Reviewing Editor has drafted this decision to help you prepare a revised submission.

In this report, Handly et al. analyzed the impact of paracrine communication on the cellular response fidelity in an epithelial cell line during wound healing. Through single cell live imaging they quantified signal and noise of Ca^2+^ and ERK responses following treatment with ATPs (which recapitulates an event happening in wound context). They first validated the system (paracrine activation of ERK by ATP) and then they analyzed the effect of differential spatial density of cell clusters on the responses. They identified that higher density correlates with lower response variability, suggesting that paracrine communication decreases cellular variability response. To quantify the observed benefit of paracrine communication they performed a computational analysis based on signal-to-noise ratio (SNR). This analysis showed that paracrine communication can increase the response SNR.

To measure the spatial wound response they developed and used a new microfluidics-based device. The single cell analysis of Ca^2+^ and ERK showed the importance of cellular position for the determination of cellular response. Applying the same computational strategy as before in a better-controlled environment, they showed again that the paracrine communication distance decreases the noise of the response. However they also identified a limit of paracrine communication, which consist in reducing the gradient magnitude.

To directly measure the paracrine communication, the authors used a co-culture system in which they can distinguish "sender" and "receiver" cells. They measured the ERK level in "receiver" cells in relation of the distance from a sender cell. The authors determined the paracrine communication distance and compared it with their predicted distance value that maximizes the SNR. Therefore they concluded that the distance between communicating cells maximize cellular response fidelity.

Essential revisions:

1) Although the new microfluidics-based device represents a great tool to study wound healing in vitro in a highly controlled manner and already provided intriguing results, it would be important to see if some of the principles of paracrine communication observed in a static media condition can also be identified in a non-static condition (Figure 2—figure supplement 1).

2) The data collected and the analysis are interesting, however they refer to an in vitro setting (this should be more clear in the Abstract). Please discuss these data in the light of in vivo data present in literature (i.e. ERK reporter has been used in vivo).

3) The analyses are largely described in the text without any equations; having the equations in the Methods or supplements would be much better, and several of the quantitative figures are described quite vaguely (Figure 1, Figure 2, and Figure 3).

4) At a number of points, the meaning of error bars and p-values are not explicitly stated, and these need to be fixed.

5) Given the setup in Figure 1, please show some quantification of ERK variability in Figures 1 and 2 This omission seems odd given that ERK is the signal that is ultimately being affected by the averaging, and which is being predicted by the analysis. Is it in fact less variable than the Ca^2+^ signal in these systems? There also seems to be a logical disconnect – the optimum communication distance shown in Figure 2 is calculated based on Ca^2+^ responses, but then compared to the ERK response in Figure 3.

6) In Figure 1 (and in general), what is the expected role of diffusion? The authors conclude that there is no upper bound on SNR as distance is increased, but their analysis doesn't appear to take diffusion into account, which would presumably limit the distance over which averaging could be effective. This should be clarified. It would also be helpful to comment on the role of the integration time over which the cellular response operates.

---

## [Author Response]

Essential revisions:1) Although the new microfluidics-based device represents a great tool to study wound healing in vitro in a highly controlled manner and already provided intriguing results, it would be important to see if some of the principles of paracrine communication observed in a static media condition can also be identified in a non-static condition (Figure 2—figure supplement 1).

We now include a section in Material and methods (“Analysis of the effect of fluid flow on paracrine communication”) with an additional figure (Figure 1—figure supplement 7) where we present an analysis of the relative contribution of diffusion (i.e. static) and advection (i.e. non-static) conditions. Our analysis shows that at physiological levels of interstitial flow, the principles of paracrine communication can also be identified.

2) The data collected and the analysis are interesting however they refer to an in vitro setting (this should be more clear in the Abstract).

We now clarify that our data was collected in an in vitro setting in the Abstract as well as within the main text (subsection “Paracrine signaling reduces response variability” and Discussion, second paragraph).

Please discuss these data in the light of in vivo data present in literature (i.e. ERK reporter has been used in vivo).

The in vitro data presented here is now discussed in light of in vivo data using the same ERK FRET reporter from 2 different papers (Hiratsuka et al. and Kumagai et al.). Please see the Discussion.

3) The analyses are largely described in the text without any equations; having the equations in the Methods or supplements would be much better, and several of the quantitative figures are described quite vaguely (Figure 1, Figure 2, and Figure 3).

We now include 14 equations to better describe the analyses used throughout the paper. Equations describing the paracrine model were added to the Methods section. In addition, multiple new sections were added to the Material and methods section including:

a) Model for paracrine communication based on local isotropic diffusion;

Signal to Noise Analysis;

b) Analysis of the effects of diffusion, secretion and integration time on paracrine communication;

c) Analysis of the effect of fluid flow on paracrine communication;

d) Analysis of the effect of cellular decoding schemes on paracrine communication.

These sections describe our model in details and include detailed analytical derivations that justify the assumption behind the model. Four new figure supplements to Figure 1 are presented to better describe the results of the additional analysis graphically. In order to better clarify Figure 1, Figure 2, and Figure 3, the figure legends were modified to indicate which analyses took place.

4) At a number of points, the meaning of error bars and p-values are not explicitly stated, and these need to be fixed.

The meaning of error bars, shaded regions, and p-values are now explicitly stated throughout the figure legends.

5) Given the setup in Figure 1, please show some quantification of ERK variability in Figures 1 and 2 This omission seems odd given that ERK is the signal that is ultimately being affected by the averaging, and which is being predicted by the analysis. Is it in fact less variable than the Ca^2+^ signal in these systems?

We added additional experimental data and additional analysis related to variability of ERK. Specifically, we added the following quantifications:

a) Figure 1—figure supplement 3 and Figure 1—figure supplement 4: Quantification of ERK variability when the paracrine signaling length-scale is manipulated;

b) Figure 2 was divided in to Figure 2 and Figure 3. Additional ERK wound response data is shown in Figure 2 and the variability of the Ca^2+^ response compared to the ERK response, shown as coefficient of variation, to wounding is shown in 2F. As expected from our model, Ca^2+^ response shows higher variability than ERK response.

Collectively the additional data demonstrates that the wound induced ERK response has lower variability than Ca^2+^ response. Because Ca^2+^ response is prior to paracrine communication and ERK response is post paracrine communication, this reduction in variability provides direct support to our paracrine communication model.

There also seems to be a logical disconnect – the optimum communication distance shown in Figure 2 is calculated based on Ca^2+^ responses, but then compared to the ERK response in Figure 3.

To prevent potential perceived logical disconnect we now clarify the difference between the Ca^2+^ response, the predicted ERK response from locally averaging the Ca^2+^ response, and the actual ERK response in both the text (subsection “Cellular response fidelity depends on the extent of paracrine signaling during wound response”) and Figure 3.

6) In Figure 1 (and in general), what is the expected role of diffusion? The authors conclude that there is no upper bound on SNR as distance is increased, but their analysis doesn't appear to take diffusion into account, which would presumably limit the distance over which averaging could be effective. This should be clarified. It would also be helpful to comment on the role of the integration time over which the cellular response operates.

We thank the reviewer for bringing up this important point. We now extended our model to explicitly take into account the effects of diffusion under conditions of limited integration time. We show an extended model that explicitly takes into account integration time in both Figures 1 and 3. The equations that are part of the derivation of this extended model are presented in Material and methods section: “Analysis of the effects of diffusion, secretion and integration time on paracrine communication”. In this section we analytically derive key expressions that show the dependency of the Paracrine Communication Distance (PCD) on the diffusion coefficient and the ratio between the number of released molecules to the sensitivity of detection. We show under what regimes integration time could become limiting and therefore extend our model to include such limitations to the potential benefit of paracrine communication.